# Vagal Nerve Activity Predicts Prognosis in Diffused Large B-Cell Lymphoma and Multiple Myeloma

**DOI:** 10.3390/jcm12030908

**Published:** 2023-01-23

**Authors:** Or Atar, Ron Ram, Irit Avivi, Odelia Amit, Roy Vitkon, Efrat Luttwak, Yael Bar-On, Yori Gidron

**Affiliations:** 1Department of Nursing, Faculty of Social Welfare and Health Sciences, Haifa University, Haifa 3498838, Israel; 2Department of Hematology, Ichilov University Hospital, Tel-Aviv 6423906, Israel

**Keywords:** vagal nerve activity, cancer prognosis, diffuse large-B cell lymphoma, multiple myeloma

## Abstract

This study examined the prognostic role of vagal nerve activity in patients with relapsed/refractory diffused large B-cell lymphoma (R/R-DLBCL) treated with chimeric antigen receptor cell therapy (CAR-T) and in patients with multiple myeloma (MM) undergoing an autologous hematopoietic cell transplantation (AutoHCT). Participants included 29 patients with R/R-DLBCL and 37 patients with MM. Inclusion criteria were: (1) age over 18; (2) diagnosed with DLBCL or MM; (3) being treated with CAR-T or AutoHCT; and (4) having an ECG prior to cell transfusion. The predictor was vagal nerve activity indexed by heart rate variability (HRV) and obtained retroactively from 10 s ECGs. The main endpoint for R/R-DLBCL was overall survival (OS), and for MM the endpoint was progression-free survival (PFS). Data of 122 patients were obtained, 66 of whom were included in the study. In DLBCL, HRV significantly predicted OS independently of confounders (e.g., performance status, disease status at cell therapy), hazard ratio (HR), and 95% confidence interval (HR = 0.20; 95%CI: 0.06–0.69). The prognostic role of disease severity was moderated by HRV: among severely disease patients, 100% died with low HRV, while only 37.5% died with high HRV. In MM, HRV significantly predicted PFS (HR = 0.19; 95%CI: 0.04–0.90) independently of confounders. Vagal nerve activity independently predicts prognosis in patients with R/R-DLBCL and with MM undergoing cell therapy. High vagal activity overrides the prognostic role of disease severity. Testing the effects of vagal nerve activation on prognosis in blood cancers is recommended.

## 1. Introduction

Diffused large B-cell lymphoma (DLBCL) is considered the most common type of non-Hodgkin’s lymphoma worldwide [1]. Most patients receive chemotherapy. However, 30–40% of patients experience disease relapse [2]. These patients often fail to achieve durable remissions with second-line therapy and therefore are considered for chimeric antigen receptor T-cell (CAR-T) therapy. The main prognostic factors predicting outcomes following CAR-T cell therapy are performance status, lactate dehydrogenase levels, and disease status [3].

Another relatively frequent hematological malignancy is multiple myeloma (MM) [4]. The standard first-line treatment in patients recently diagnosed with MM is high-dose chemotherapy with melphalan followed by autologous hematopoietic stem cell transplantation (AutoHCT) [5]. The main prognostic factors predicting outcomes following AutoHCT include MM-related cytogenetic factors and disease status at transplantation [6].

Despite progress in treating DLBCL and MM [7,8], the current 5-year overall survival rates are 60–70% [9] and 67% [10], respectively. One way to possibly make progress is by identifying a prognostic protective factor which also inhibits multiple biological factors that enhance cancer progression. One such candidate protective factor might be the vagal nerve.

The vagus is a major branch of the parasympathetic nervous system. Vagal activity can be non-invasively indexed by heart rate variability (HRV), i.e., fluctuations in the interval between normal heart beats. A previous study showed that there is a robust correlation between HRV and actual activity of the vagal nerve itself (r = 0.88; [11]).

It has been hypothesized [12] that efferent vagal activity, indexed by high HRV, is related to slower tumor progression because it inhibits inflammation [13], oxidative stress [14], and excessive sympathetic activity [15], three underlying mechanisms contributing to the onset and progression of cancer [16,17,18]. This hypothesis has been confirmed only in relation to inflammation. De Couck et al. [19] found that inflammation (C-reactive protein) statistically mediated the association between higher HRV and prolonged survival in pancreatic cancer.

Moreover, epidemiological evidence shows that higher HRV was found to be an independent predictor of lower tumor marker levels [20,21] and prolonged survival in several types of solid cancers (e.g., lung, breast, pancreatic) as seen in three reviews [22,23,24]. Although the predictive value of vagal nerve activity has been confirmed in various solid cancers, its value in the prediction of prognosis in hematologic cancer patients remains relatively unknown.

As far as we know, only one study examined the relationship between HRV and prognosis in hematologic cancer and found that increased probability of survival was related to initially higher levels of HRV [25]. Yet, that study was based on a small (*N* = 27) and heterogenic sample, limited to patients with insomnia, and confounders (e.g., disease severity, treatments) were not included in the statistical analyses.

The three hypothesized mechanisms (sympathetic activity, inflammation, and oxidative stress) mentioned above also play a role in blood cancers. Sympathetic activity increases the proliferation of myeloid cell development [26], influences cancer pathogenesis via regulation of the bone marrow environment [27], impairs the response to immunotherapy in B-cell lymphoma [28], and might have a central role in MM cells’ progression [29].

Concerning inflammation, high levels of interleukin-6 were associated with aggressive disease and low survival rates in both MM [30] and DLBCL [31]. Similarly, oxidative stress markers have been shown to have prognostic value and to be associated with adverse outcomes in patients with DLBCL [32,33] and to be increased in MM [34].

An additional mechanism that has not yet been investigated in the context of the vagus and cancer may involve lactate dehydrogenase (LDH). LDH is a cellular enzyme that plays an important role in cellular respiration and in cell proliferation [35] and is a marker of cell death. Furthermore, LDH is a well-known prognostic factor in DLBCL [36] and MM [37]. Vagal nerve activity may reduce LDH since the vagus can inhibit ischemia [38], which thus prevents cell death. Indeed, one study found in the context of ischemia that vagal nerve stimulation decreases LDH [39]. Together, vagal activity may predict a better prognosis in these hematologic cancers as well because this nerve inhibits sympathetic hyperactivity, inflammation and LDH. In the present study we focus on inflammation and LDH as possible mechanisms.

The purpose of this study was to examine the prognostic role of HRV in human lymphoma and myeloma while considering several confounders. We focused on two relatively homogenous groups of patients—1. patients with relapse/refractory DLBCL (R/R-DLBCL) who were admitted to anti-CD19 chimeric antigen receptor T cell (CAR-T) therapy and 2. patients with newly diagnosed MM who were admitted after 3–4 courses of induction line to AutoHCT. We hypothesized that (1) high baseline HRV would predict longer overall survival (OS) in R/R-DLBCL and longer progression-free survival (PFS) in MM independent of confounders (e.g., age, treatments), and (2) lower levels of inflammation and LDH would mediate these relationships.

## 2. Methods

### 2.1. Design

This was a single-center retrospective analysis at the BMT Unit, Tel Aviv Sourasky Medical Center. Consecutive patients with MM and R/R-DLBCL undergoing cellular therapy were identified by performing a search in a computerized system.

We obtained existing archival data on patients’ medical backgrounds and HRV and then examined the prospective relationship between these parameters measured prior to transplantation and patients’ OS for R/R-DLBCL and PFS for MM. These two outcomes were chosen due to the number of cases in each disease. Specifically, in MM, most patients fortunately survived, and thus PFS was chosen, while in DLBCL, unfortunately, 53% of patients died, and thus OS was chosen.

### 2.2. Patient Cohorts

All patients (R/R-DLBCL, *N* = 29 and MM, *N* = 37) were treated between 2019 and 2021. Inclusion criteria for both samples were patients who (1) were above the age of 18; (2) were diagnosed with either DLBCL or MM; (3) were candidates for undergoing anti-CD-19 CAR-T cell in patients with RR-DLBCL or underwent autologous HCT in patients with MM; and (4) underwent a 10 s ECG prior to cell therapy. Exclusion criteria were chronic or acute inflammatory disease that may affect HRV.

These sample sizes are justified as follows: based on the HRV data of Schebier et al., 2018, on hematological cancers, and assuming a smaller SD of HRV by recruiting more homogenous samples in the present study, and assuming a 40% difference in SDNN between alive and dead patients [19] and a statistical power of 0.8 and statistical significance of 0.05, 38 patients were expected to be sufficient.

### 2.3. Measures

Confounders and mediators. These included age, gender, number of prior treatments, status of disease at cell therapy, additional comorbidities (hypertension, renal failure, heart failure, atrial fibrillation, coronary artery disease, and diabetes mellitus), body mass index (BMI), and performance status (ECOG for R/R-DLBCL and Karnofsky for MM). We obtained from patients’ medical records levels of C-reactive protein (CRP) and lactate dehydrogenase (LDH) before CAR-T cell therapy or transplantation (at the same day as the ECG).

Though other prognostic indexes of disease severity could be used, such as FISH, IPI, and ISS, we considered disease status in each sample alone because it reflected the actual disease severity at the time of obtaining the ECG from which vagal activity was determined. Furthermore, for some of the other prognostic indexes, data were missing.

Vagal Nerve Activity. Heart rate variability (HRV), the index of vagal nerve activity, was derived from patients’ 10 s ECG before CAR-T cell therapy or transplantation. HRV was calculated based on the standard deviation of patients’ R–R intervals (SDNN), in msec, and the root mean square of successive differences (RMSSD) between consecutive R–R intervals. Such brief HRV provided prognostic value in various cancers in past studies [19] and correlates with ECGs of longer durations [40].

Outcomes. For R/R-DLBCL, OS was the main outcome, and for MM, PFS was the main outcome. The follow-up period for both samples was 1 year.

### 2.4. Statistical Analysis

The results are presented separately for R/R-DLBCL (Sample 1) and MM (Sample 2). Categorical variables are described with contingency tables, including frequencies and percentages. Continuous variables are described as mean, median, and standard deviation. Most variables are considered as continuous unless otherwise specified. Confidence intervals were calculated for survival analyses at the (2-sided) 95% level of confidence. A 2-sided *p* value of <0.05 was considered to be statistically significant.

We first examined every predictor alone in relation to OS or PFS using a Cox regression. Finally, a multivariate Cox regression analysis was performed in relation to OS or PFS while also considering the time until death or progression or the time until censorship date for alive patients or non-progressing patients. Confounders which significantly predicted each outcome were entered into the regressions. For the continuous biological variables, in case they were not normally distributed, we log transformed the original data.

CRP and LDH were considered as possible explanatory mediators rather than simple predictors, which could explain the relationship between HRV and the clinical outcomes. To determine statistically the mediating role of CRP and LDH, we performed the following tests. We first examined the direct relationship between CRP and LDH separately with OS or PFS, using Cox regression. Then we examined the relationship between CRP and LDH with HRV by Pearson correlation. Finally, we entered CRP and LDH separately into the multivariate Cox regression with all other predictors and HRV. If the predictive role of HRV in relation to the clinical outcomes diminished or was no longer significant after adding CRP or LDH, then we inferred that that variable was the mediator.

Finally, we examined whether HRV moderated the prognostic role of well-known predictors. Specifically, we tested whether disease status predicted prognosis separately in patients with low and high HRV.

## 3. Results

### 3.1. Sample 1—R/R-DLBCL

#### 3.1.1. Descriptive Statistics

As we can see in Table 1, patients’ mean age was below 70, their average BMI was overweight, the CRP was excessive, and their mean SDNN was low and below the median of cancer patients [41]. Approximately half of the sample had a progressive disease, one-third had hypertension or diabetes, and 55% died. Five patients died before receiving CAR-T cell therapy. The median survival time was 177 days.

#### 3.1.2. Predictors of Survival in R/R-DLBCL

We then examined each predictor alone in relation to OS. Among all variables, only the number of past treatments (H.R = 2.28, *p* = 0.01), disease status (H.R = 1.83, *p* = 0.07), ECOG (H.R = 2.51, *p* = 0.01), Hb (H.R = 0.63, *p* = 0.03), and SDNN (H.R = 0.33, *p* = 0.03) were significant predictors of OS. These were then included in the multivariate cox regression (see Table 2). All other confounders did not significantly predict OS. The results did not change when excluding the five patients who died before receiving CAR-T cell therapy. To avoid reducing statistical power further, we included them in all analyses.

In a multivariate Cox regression, SDNN significantly predicted survival, independently of the other confounders. Patients with high HRV had 4.85 times greater chances of surviving than patients with low HRV (See Figure 1).

#### 3.1.3. Mediation by CRP and LDH

We then examined the mediating roles of CRP and LDH in the relationship between HRV and survival. HRV significantly and negatively correlated with CRP (r = −0.63, *p* < 0.01) and with LDH (r = −0.55, *p* < 0.01). Both CRP (H.R = 1.01, *p* < 0.01) and LDH (H.R = 1.00, *p* < 0.01) also significantly predicted survival. Thus, we added each separately to the multivariate Cox regression previously shown in Table 2. When adding CRP to the regression, HRV only tended to significantly predict survival (*p* = 0.06). In contrast, when adding LDH to the regression, HRV no longer significantly predicted survival (*p* = 0.47). These results show that LDH mediated the relationship between HRV and survival.

#### 3.1.4. Moderation by Vagal Nerve Activity

For this analysis, we divided the sample into high and low HRV at the cutoff of 12 milliseconds of SDNN, which was the median in that sample. Similarly, disease status was categorized at approximate median number of cases (progressive disease versus all other levels). Disease status predicted death only in patients with low HRV (X^2^(1) = 9.24, *p* < 0.01) but not in patients with high HRV (X^2^(1) = 0.04, *p* = 0.83). Importantly, among patients with severe disease, 100% of them died when HRV was low, while only 37.5% of severely diseased patients died when HRV was high. Similarly, ECOG significantly predicted survival only in patients with low HRV (*p* = 0.05), but not in patients with high HRV (*p* = 0.22). Thus, high HRV moderated (weakened) the prognostic roles of disease status and ECOG. 

### 3.2. Sample 2—MM

#### 3.2.1. Descriptive Statistics

As we can see in Table 3, patients’ mean age was below 70, two-thirds were men, their average BMI was overweight, and their mean SDNN was low and below the median of cancer patients [41]. Twenty percent of the sample had a progressive disease at cell therapy. Of the entire sample, 6.1% died (see Table 3). The median PFS time was 302 days.

#### 3.2.2. Predictors of Progression in MM

We then examined each predictor alone in relation to PFS. Among all variables, disease severity (H.R = 1.86, *p* < 0.01) and RMSSD (H.R = 0.26, *p* < 0.05) were significant predictors of PFS. In this sample, SDNN did not predict progression (*p* > 0.05). Thus, to be more rigorous, RMSSD and disease severity were then included in the multivariate Cox regression as well as prior treatments (see Table 4).

In a multivariate Cox regression, RMSSD significantly predicted PFS independently of the other confounders. Patients with high HRV had 5.23 times greater chances of PFS than patients with low HRV (See Table 4 and Figure 2).

#### 3.2.3. Mediation by CRP and LDH

Since LDH and CRP were not correlated with RMSSD, their role as mediators was not examined. However, each of them predicted PFS alone (H.R = 1.00, *p* = 0.04 for LDH; H.R = 1.03, *p* = 0.02 for CRP).

#### 3.2.4. Moderation by Vagal Activity

For this analysis, we divided the sample into high and low HRV at the cutoff of 30 milliseconds of RMSSD, which was the median. Similarly, disease status was categorized at approximate median number of cases (progressive disease and partial remission versus complete remission and VGPR). Disease status predicted PFS only in patients with low HRV (X^2^(1) = 10.5, *p* = 0.001) but not in patients with high HRV (X^2^(1) = 0.80, *p* > 0.05). Importantly, among patients with severe disease, 100% of them progressed when HRV was low, while only 33.3% of patients with severe disease progressed when HRV was high. Such moderation was not found for Karnofsky.

## 4. Discussion

The purpose of this study was to examine the independent prognostic role of the vagus nerve (indexed by HRV) in RR-DLBCL and MM. In RR-DLBCL, we found that HRV (SDNN) was a predictor of survival independent of confounders. Patients with higher HRV had 4.85 times greater chances of surviving compared to those with low HRV independent of multiple confounders, including disease severity. Moreover, this relation was mediated by LDH and CRP levels. Finally, we found that high HRV moderated (weakened) the prognostic roles of disease severity and ECOG in relation to survival. ECOG and disease severity predicted survival only in patients with low HRV, while this did not occur in patients with high HRV.

For MM, we found that HRV (RMSSD) was a predictor of PFS independent of confounders. Patients with higher HRV had 5.23 times greater chances of not progressing compared to those with low HRV. However, LDH and CRP did not mediate this relationship in MM. Finally, we found again that high HRV moderated (weakened) the prognostic role of disease severity in relation to disease progression. Disease severity predicted PFS only in MM patients with low but not high HRV.

These results demonstrate the consistency of the prognostic role of the vagus nerve in two different hematologic cancers. These results are in line with Scheiber et al. [25], who found that increases in HRV after transplantation predicted a higher chance of survival. In that study, HRV was measured twice (before and after transplantation), enabling them to examine the prognostic value of change in HRV. In contrast, in the present study, HRV was only measured once. However, that study was based on a small and heterogenic sample, limited to patients with insomnia, and confounders (e.g., disease severity, treatments) were not included in the statistical analyses. In the present study, both our samples were homogeneous, and the role of confounders was considered in multivariate analyses.

Moreover, the results of the present study are in line with previous studies showing the prognostic role of HRV in solid cancers, such as colon, breast, pancreas, and liver cancer [19,20,42], and in line with systematic reviews and a meta-analysis showing the independent prognostic role of HRV in cancer [22,23,24]. Our results extend these results to two types of blood cancer, which raises the hypothesis that the vagus may affect blood cancer prognosis through a systemic mechanism.

Indeed, in the R/R-DLBCL sample, both LDH and CRP partly mediated the relationship between HRV and survival. The finding concerning CRP replicates those of De Couck et al. [19] showing that CRP mediated the relationship between HRV and survival in pancreatic cancer. The mechanism underlying these results was originally hypothesized by Gidron et al. [12], stating that the vagus inhibits inflammation [43,44], which otherwise contributes to tumorigenesis both in solid [45] and blood cancers [30,31].

The vagus might inhibit inflammation through two mechanisms. First, by translation of peripheral IL-1 signals to acetylcholine, which activates the hypothalamic–pituitary–adrenal axis, resulting in cortisol, which inhibits inflammation [44]. Second, by translation of descending cholinergic to adrenergic signals which enter the spleen, where they bind to a sub-type of T cells which then secrete acetylcholine. The later binds to its receptor on splenic macrophages, which suppresses the synthesis of inflammatory signals [43].

Regarding LDH, as far as we know, the precise mechanism by which the vagus affects cancer prognosis via LDH is unknown. As mentioned above, LDH rises after cell death, which occurs also due to ischemia. In contrast, vagal nerve stimulation reduced LDH in the context of ischemia [39]. The vagus nerve negates ischemia via vasodilatation in addition to inhibiting sympathetically induced vasoconstriction. Since LDH predicts poor prognosis in blood cancers [36,37], the anti-ischemic mechanism of vagal activation might partly explain the mediating role of LDH observed in the present study. However, as mentioned before, we did not find the mediating role of CRP and LDH in relation to progression in MM. Future studies should examine these mediators more deeply.

Importantly, in both samples, HRV moderated the respective prognostic roles of disease severity in relation to OS and PFS. As mentioned above, only in patients with low HRV, all of those with severe disease died or progressed. In contrast, in patients with HRV, far fewer patients died or progressed in the disease despite having a severe disease stage. These findings are in line with a study showing similar moderation in relation to tumor markers in colon and prostate cancer [46]. Though based on correlational designs only, these results might strongly demonstrate the prognostic and protective roles of high vagal nerve activity in hematological cancers.

The present study had several limitations. First, both samples were relatively small. On one hand, this could reduce the generalizability of our findings. However, our samples were relatively homogeneous, and confounders were controlled for in the analyses. Second, the HRV was measured retroactively in a 10 s ECG, which did not enable us to derive frequency/domain HRV parameters such as high frequency HRV. Nevertheless, several studies showed that such brief HRV measures predicted survival in the population [47] and predict cancer prognosis [19,42]. This retrospective design limited our control over the measures of confounders, which could have prognostic value (e.g., physical activity). Third, the time between performing the ECG and transplantation varied between patients, which could affect HRV or other confounder levels. Finally, the follow-up periods, particularly for MM patients, were relatively short, considering that they are expected to progress within a longer period of time.

Future studies should address these limitations by conducting prospective studies on larger samples and measuring HRV several times and in consistent intervals for all patients. Despite these limitations, the consistency of the results presented above, the controlling for confounders, and the use of homogenous samples demonstrate the robust independent prognostic role of HRV in blood cancers. Future studies should examine the causal and therapeutic roles of the vagus nerve in blood cancers by performing a randomized, controlled trial. Such a trial can include vagal activation by electric stimulation, biofeedback, medications, etc.

## Figures and Tables

**Figure 1 jcm-12-00908-f001:**
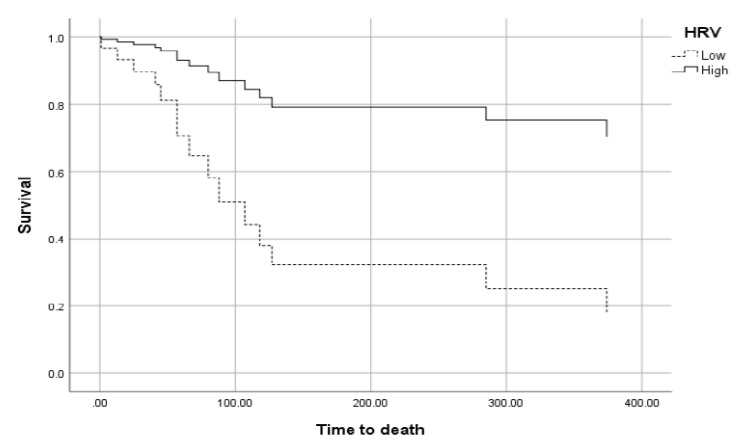
The relationship between HRV and overall survival in RR-DLBCL.

**Figure 2 jcm-12-00908-f002:**
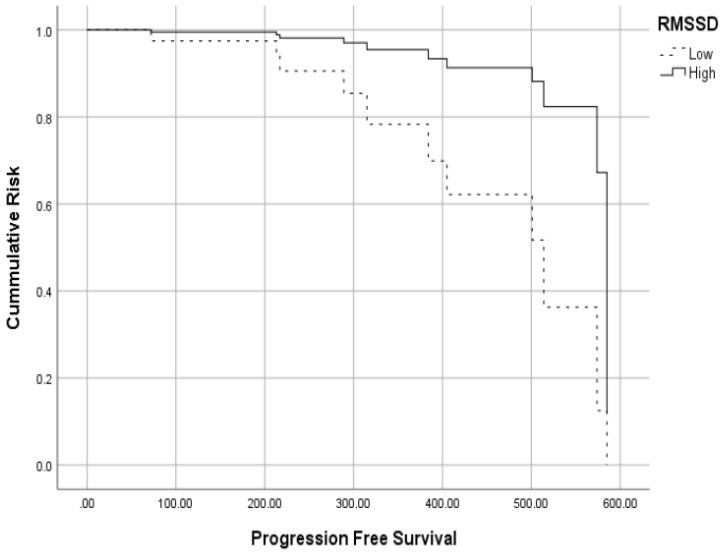
The relationship between HRV and PFS in MM sample.

**Table 1 jcm-12-00908-t001:** Relapsed/refractory diffused large B-cell lymphoma sample.

A. Continuous Variables Variable	Mean	SD
Age	67.33	13.99
BMI	26.11	4.31
ECOG	2.13	0.78
Hb	9.10	1.52
LDH	667.13	734.82
CRP	55.97	53.54
SDNN	14.58	11.91
RMSSD	50.71	35.28
Time to death (days)	221.65	180.77
**B. Categorical Variables Variable %**		
Gender		
Men	44.8%	
Women	55.2%	
Disease Status at Cell Therapy *		
1. Complete Remission	12.5%	
2. Partial Remission	15.6%	
3. Stable Disease	15.6%	
4. Progressive Disease	56.3%	
Coronary Heart Disease		
Yes	10.3%	
No	89.7%	
Atrial Fibrillation		
Yes	13.8%	
No	86.2%	
Chronic Heart failure		
Yes	14.3%	
No	85.7%	
Renal Failure		
Yes	17.2%	
No	82.8%	
Hypertension		
Yes	37.9%	
No	62.1%	
Death		
Yes	55.2%	
No	44.8%	

Note: SDNN = standard deviation of R–R intervals; RMSSD = root mean square of successive differences between proximal R–R intervals; Hb = hemoglobin; CRP = C-reactive protein; LDH = lactate dehydrogenase; * = before transplantation.

**Table 2 jcm-12-00908-t002:** Multivariate Cox regression between all significant bivariate predictors and overall survival in R/R-DLBCL.

Predictor	β	SE	Sig.	H.R	95% CI
Past treatments	1.01	0.45	0.02	2.76	1.14–6.72
Status of disease	1.64	0.74	0.02	5.17	1.20–22.28
ECOG	−0.075	0.51	0.88	0.92	0.33–2.53
Hb	−0.44	0.30	0.14	0.64	0.35–1.16
SDNN	−1.58	0.62	0.01	0.20	0.06–0.69

Note: SDNN = standard deviation of R–R intervals; SE = standard error of the mean; Sig. = significance; H.R = hazard ratio; CI = 95% confidence interval. R/R-DLBCL = relapse/refractory diffused large B-cell lymphoma.

**Table 3 jcm-12-00908-t003:** Study characteristics of multiple myeloma sample.

A. Continuous Variables Variable	Mean	SD
Age	61.40	7.02
BMI	26.79	3.32
Karnofsky	88.64	11.09
Hb	11.37	1.66
LDH	427.24	272.59
CRP	15.62	26.09
SDNN	14.60	10.63
RMSSD	47.62	36.03
Progression-free survival (days)	353	160
**B. Categorical Variables Variable %**		
Gender		
Men	67.6%	
Women	32.4%	
Disease Status at Cell Therapy *		
1. Complete Remission	14.3%	
2. VGPR	48.6%	
3. Partial Remission	17.1%	
4. Progressive Disease	20.0%	
Coronary Heart Disease		
Yes	2.7%	
No	97.3%	
Renal Failure		
Yes	2.7%	
No	97.3%	
Hypertension		
Yes	29.7%	
No	70.3%	
Diabetes		
Yes	13.5%	
No	86.5%	
Death		
Yes	6.1%	
No	93.9%	

Note: SDNN = standard deviation of R–R intervals; RMSSD = root mean square of successive differences between proximal R–R intervals; Hb = hemoglobin; CRP = C-reactive protein; LDH = lactate dehydrogenase; * = before transplantation.

**Table 4 jcm-12-00908-t004:** Multivariate Cox regression between all significant bivariate predictors and PFS.

Predictor	β	SE	Sig.	H.R	95% CI
Number of prior treatments	0.08	0.36	0.81	1.08	0.53–2.24
Status of disease	2.49	0.88	0.00	12.08	2.13–68.53
RMSSD	−1.65	0.79	0.03	0.191	0.04–0.90

Note: RMSSD = root mean square of successive differences between intervals; SE = standard error of the mean; Sig. = significance; H.R = hazard ratio; 95% CI = 95% confidence interval.

## Data Availability

The data will be shared upon request via the corresponding author.

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
