# Peer review of "Vagal Nerve Activity Predicts Prognosis in Diffused Large B-Cell Lymphoma and Multiple Myeloma"

_jcm, 2023, doi:10.3390/jcm12030908_

Round 1

Reviewer 1 Report

In their manuscript, Atar et al. investigated the prognostic value of vagus nerve activity in patients undergoing conventional oncological treatment for B-cell lymphoma or multiple myeloma. Vagal activity was assessed retrospectively by determining heart rate variability from 10-second ECG recordings. The authors showed that vagal activity independently predicted prognosis in both groups of cancer patients. Importantly, cancer patients with high vagal activity (reflected by high heart rate variability) have been shown to have a significantly better prognosis. These impressive data suggest that heart rate variability may represent an important prognostic marker and that approaches to increase heart rate variability may be useful in the treatment of blood cancers.

Comments:

I recommend changing the title to: Vagal nerve activity predicts prognosis in diffused large B-cell lymphoma and multiple myeloma

Page 2: “multiple biological factors that enhance tumorigenesis.” Please consider changing the term "tumorigenesis" to "cancer progression" because tumorigenesis is associated with prevention, cancer progression with prognosis.

Page 2: Provide a citation for the statement "vagus can inhibit ischemia". Which mechanism is responsible? Please note that the vagal innervation of vessels is limited to only a few vascular beds.

Page 3: Rat heart variability is affected by several factors - was the ECG measurement performed under standard conditions?

Results: It is necessary to note that hypertension, diabetes, renal failure and obesity may be characterized in general by increased sympathetic nerve activity and decreased vagal activity.

Why was measurement of heart rat variability not also performed after the treatment?

Discussion:

Page 9: Please change “hypo-thalamic-pituitary-adrenal axis” to “hypothalamic-pituitary-adrenal axis”

Page 9/10: Vasodilatation mediated by the vagus nerve is limited to only a few blood vessels (e.g. genitals). The vasodilatory effect can be explained by a partial reciprocity between the activities of the vagus and sympathetic nerves. This means that if vagal activity predominates, then sympathetic nerve activity (or its effect on effector cells) may be reduced, resulting in vasodilation.

Page 10: As noted by authors, the 10-second ECG recording may be a limitation of the study.

Author Response

Thank you for the comments of reviewer 1.

  1. We changed as suggested.
  2. Done. 
  3. We added a citation for the role of the vagus in vasodilatation.
  4.  Indeed, we will address the issue in the limitations. 
  5. We agree with this comment. However, we added to the multivariate regression any variable and co-morbidity, which predicted survival.
  6. This was a retroactive study, and no ECG after treatment was available. 
  7. We added to the discussion the role of the reduction of sympathetic activation in vasodilatation in addition to vagal activation. 

Reviewer 2 Report

I have read the manuscript numbered “jcm-2157599” and entitled “Does vagal nerve activity predict prognosis in diffused large B-cell lymphoma and multiple myeloma?”. The subject is very interesting, and I believe it carries an original hypothesis. The authors here included a total of 66 patients and assessed vagal nerve activity via heart rate variability.  

·         The introduction is clear and addresses the current literature and what the current study adds.

·         The authors may mention why they choose patients with RR DLBCL before CAR-T or newly diagnosed MM patients. Why not all RR DLBCL or RR MM? If there is a specific reason it should be mentioned.

·         Method section is clear and structured. The statistical section is well-written with details. Which software and version/package used should be mentioned.

·         I had difficulty in understanding table 2. It seems that the same data was presented twice.  It should be checked or explained.

·         In Figures 1 and 2, the duration in the X axis should be presented (days, months, etc.) and the decimals can be omitted.

·         Although the results are structured, tables and figures were utilized, I found the results hard to digest. It could be modified to a more reader-friendly version. Maybe more tables less text or shorter direct language can be utilized.

·         As the authors indicated, including more than one-time point of HRV measurement, such as remission and active disease states in the same patient, could really strengthen the study.

·         The discussion is adequate. The authors mentioned the main findings and compared the study with current literature, mentioning the limitations and future directions. 

Author Response

Thank you for the comments of reviewers 2. 

  1. We had difficulty recruiting sufficient patients for each diagnosis. Thus, we included 2 different blood cancers. However, the consistency in the findings might suggest the applicability and reliability of the role of HRV in the prognosis of blood cancers.
  2. We checked, and there is no repetition in table 2. Perhaps the reviewer received a version of the manuscript with a technical error. 
  3. With the permission of the reviewer, we kept the results in the original format, but we clarified the language.